# Effect of the Crohn’s Disease Exclusion Diet (CDED) on the Fecal Calprotectin Level in Children with Active Crohn’s Disease

**DOI:** 10.3390/jcm11144146

**Published:** 2022-07-17

**Authors:** Małgorzata Matuszczyk, Monika Meglicka, Anna Wiernicka, Dorota Jarzębicka, Marcin Osiecki, Marta Kotkowicz-Szczur, Jarosław Kierkuś

**Affiliations:** Department of Gastroenterology, Hepatology, Feeding Disorders and Pediatrics, The Children’s Memorial Health Institute, 04-730 Warsaw, Poland; m.meglicka@wip.waw.pl (M.M.); a.wiernicka@ipczd.pl (A.W.); d.jarzebicka@wip.waw.pl (D.J.); m.osiecki@ipczd.pl (M.O.); m.kotkowicz@wip.waw.pl (M.K.-S.)

**Keywords:** Crohn’s disease, nutritional therapy, children, fecal calprotectin, remission

## Abstract

(1) Background: The CDED + PEN (partial enteral nutrition) is a promising method of nutritional treatment in active Crohn’s disease (CD). An increase in fecal calprotectin (FCP) level—a marker of mucosal inflammation—happens to be the first evidence of Crohn’s disease exacerbation that appears ahead of clinical symptoms and usually co-exists with them. In this study, we present our own experience with using the CDED + PEN in the treatment of children with CD and an increased FCP level. (2) Methods: In total, 48 children (male/female: 27/21) aged 4–17 years (median value = 13.43; IQR = 4.00) were treated with CDED + PEN between June 2019 and July 2021. The main inclusion criteria for the study was active CD defined as an FCP level ≥ 250.00 µg/g. Patients with severe clinical manifestation of CD (PCDAI >40.00), as well as ones who started any new concomitant CD treatment later than at least 4 weeks before the start of dietary intervention, were excluded from the analysis. The PCDAI and fecal calprotectin level were assessed at weeks 0 and 12. The primary endpoint was ITT normalization of FCP level, i.e., a result < 250.00 µg/g at week 12. The Wilcoxon Matched Pairs Test was used for statistical analysis. (3) Results: The normalization of the FCP level was obtained in 17 children (35.42%) and an FCP level decrease of at least 50% occurred in 26 patients (54.17%). The reduction in fecal calprotectin level between week 0 and week 12 was statistically significant with a median value of 1045.00 µg/g; IQR = 1188.00, and 363.00 µg/g; IQR = 665.00, respectively (*p* < 0.05). Among 29 patients who were not in clinical remission at baseline, 16 (55.17%) achieved clinical remission (PCDAI < 10.00) at week 12 and 20 (68.97%) obtained a clinical response defined as at least a 12.50 point drop in PCDAI or remission. In this group, the reduction in PCDAI between baseline and week 12 was statistically significant (median value = 20.00 points; IQR = 7.50 and 5.00 points; IQR = 5.00, respectively (*p* < 0.05)). All patients with a normal FCP level at week 12 were in clinical remission and 16 (94.13%) of them had a normal CRP (C-reactive protein) value. In 10 children (20.83%) the full course of 12 weeks with CDED + PEN was not completed or the concomitant therapy had been started before week 12 due to the lack of efficacy/intolerance of nutritional treatment. (4) Conclusions: The 12-week course of treatment with the CDED + PEN has a beneficial effect on the fecal calprotectin level in children with active CD. The dietary intervention led to a significant decrease in the FCP level in the studied group and to the normalization of this parameter in every third patient.

## 1. Introduction

Recent decades have seen a dramatic increase in the incidence of Crohn’s disease (CD)—an incurable inflammatory disease of the gastrointestinal tract [1]. This trend is observed both among adults and in the pediatric population [2]. At present, the etiology of the disease has not been clearly established and, therefore, no causal treatment is available. The therapy aims to achieve a deep remission, with the intestinal mucosa healed [3]. Years of experience have shown that the effectiveness of conventional treatment (e.g., the use of corticosteroids or immunosuppressants) is not satisfactory and its chronic use is associated with a high risk of adverse effects [4]. The modern and safer biological therapy does not meet the needs of all patients [5,6,7,8]. Recurrent nature of CD remains typical, with many patients experiencing resistance/loss of response to subsequent drugs and some eventually requiring surgical intervention [9,10,11].

On the other hand, recent findings suggest that external environmental factors, including diet, play an essential role in the pathomechanism of Crohn’s disease [12]. Hence, nutritional therapy is gaining an increasing number of supporters among professionals. Patients, discouraged by the lack of effectiveness or side effects of pharmacological treatment, are also increasingly searching for such an alternative. It is known today that, as a result of certain dietary factors, the delicate immunological balance between the microbiota and the intestinal mucosa can be disrupted, resulting in chronic inflammation [13]. This hypothesis is supported by a number of previous publications that showed a correlation between the so-called Western (industrialized) dietary pattern and an increased risk of CD [14,15,16,17,18,19,20,21,22].

Nutritional therapy for Crohn’s disease has been used for years. However, until recently, the only nutritional therapy with proven therapeutic efficacy has been exclusive enteral nutrition (EEN), which has been recommended as the first-line treatment for active Crohn’s disease in children. Numerous studies have demonstrated that, in the pediatric population, EEN and glicocorticosteroids have comparable therapeutic efficacy, with a significantly better safety profile of nutritional therapy and a more favorable effect on mucosal healing and nutritional status [11,23,24,25,26]. Unfortunately, EEN is a therapy that is not without its drawbacks, which has significantly limited its potential for widespread use [27,28,29]. Over 10 years ago, Professor Arie Levine (Wolfson Medical Center, Tel Aviv) developed an innovative method of nutritional therapy—the Crohn’s Disease Exclusion Diet (CDED). The first results of observational studies showing promising effects of its use in combination with partial enteral nutrition (PEN) were presented in 2014 [30] and 2017 [31]. In 2019, landmark results from a multicenter randomized trial were published showing that CDED + PEN and standard treatment (EEN) were comparably effective in inducing clinical remission in children with exacerbations of CD, and therapy tolerance was significantly higher in the intervention group. In addition, in the group treated with the CDED + PEN, in contrast to the control group, a reduction in intestinal permeability and the beneficial changes in the composition of microbiota [32] and in the metabolome [33] were shown, which may be important in the long-term maintenance of treatment effects. In the last year, the high efficacy of such nutritional therapy has been confirmed in the adult population [34,35].

Despite encouraging results, CDED + PEN has not yet found its place as a treatment for Crohn’s disease and further studies are needed. In previous publications, the primary parameter for assessing the efficacy of the CDED + PEN in the treatment of patients with active CD was the clinical disease activity index scales (CDAI in adults and PCDAI in children), in which some of the components are subjective assessments of disease symptoms. In the present study, the fecal calprotectin level, a recognized and non-invasive marker of intestinal inflammation, was used as the main inclusion criteria and primary endpoint [25]. The authors believe that such an analysis will allow a more objective assessment of the effectiveness of nutritional therapy and will be an important addition to the already published results.

## 2. Materials and Methods

The analysis included 48 children diagnosed with Crohn’s disease. The main inclusion criterion was active Crohn’s disease, defined as a fecal calprotectin level ≥ 250.00 µg/g. Patients with a severe clinical disease manifestation (PCDAI > 40.00), and in whom any new treatment for Crohn’s disease was implemented or the dose of existing medication was modified later than at least 4 weeks before the start of the nutritional intervention were excluded from the analysis.

The baseline characteristics of the studied group are shown in Table 1.

The age of patients at the beginning of nutritional therapy ranged from 4 to 17 years, with a median of 12 years. The distribution in terms of the sexes was comparable. Based on the assessment of clinical symptoms (i.e., PCDAI score), most of the patients had mild to moderate exacerbations of Crohn’s disease, but one-third of them were in clinical remission at baseline. The distribution between children with a new diagnosis of the disease and those with a subsequent exacerbation was comparable. At baseline, 27.08% of patients were found to be malnourished or underweight based on their BMI score according to the Polish percentile charts.

Nutritional therapy according to rules of the CDED + PEN was implemented in all patients between June 2019 and July 2021. The protocol of CDED + PEN is divided into three phases—including two (lasting 6 weeks each) in the induction stage and a third, the maintenance phase, which should be continued for a minimum of 9 months and optimally considered as the target diet. This study analyzed the therapeutic effects obtained only after the induction stage, i.e., after 12 weeks of nutritional therapy.

According to the principles of the CDED + PEN, within the natural foods that patients could consume during the induction stage, there was a division into obligatory products, i.e., recommended for daily consumption (to ensure adequate nutritional value of the diet and substrates for the production of short-chain fatty acids); neutral (which were intended to add variety to the daily diet but were not required to be consumed daily) and prohibited. In phase 1, due to the most significant restrictions on the products allowed to be eaten, 50% of the daily energy requirements were met with an enteral diet. Phase 2 was a time of re-exposure to selected products that had to be eliminated for the first 6 weeks. Due to the increasing variety of foods allowed, the recommended percentage of the energy provided as formula at this stage has decreased to 25% of the requirement. Meeting part of the nutritional needs with a complete enteral diet was aimed at maintaining/improving nutritional status and preventing nutritional deficiencies during the period of most severe nutritional restriction. The principles of formula selection were the same as those for exclusive enteral nutrition. The standard recommendation was a polymeric, normocaloric (1 kcal/1 mL) formula for patients with Crohn’s disease, i.e., Modulen IBD (Nestlé, Vevey, Switzerland). Exceptions were patients with a history of intolerance to this enteral diet or a confirmed allergy to cow’s milk protein, for whom a hydrolyzed protein formula was recommended as an alternative.

The study included 3 visits—baseline (at week 0) and at the end of phase 1 (i.e., after 6 weeks) and phase 2 (i.e., after 12 weeks) of the nutritional therapy. At each visit, nutritional status (weight, body length, and BMI) and clinical disease activity (PCDAI scale) were assessed and inflammatory parameters (CRP, ESR and fecal calprotectin level) were analyzed. All patients received ongoing supervision from a physician and a dietician trained in the rules of the nutritional intervention used. At the baseline visit, the current energy requirements and the resulting recommended supply of enteral diet and obligatory products in the first phase of nutritional therapy were determined depending on age, gender, and nutritional status. All patients and their parents were trained in the rules of the CDED + PEN and received detailed written recommendations, as well as access to the ModuLife application (an application with a database of recipes and menus adapted to each phase of the diet). Ongoing contact with a dietician was provided for them if they had any questions/doubts about following the nutritional treatment. At visits 2 and 3, in addition to the medical consultation after which a decision was made whether to continue the CDED + PEN or to modify the therapy, another dietary consultation was held to assess tolerance and adherence to the diet and to discuss dietary guidelines appropriate for the next phase of treatment. In addition, at visit 2, based on the interview with the patient and on the result of the assessment of nutritional status, the recommended amounts of enteral formula and obligatory products were modified, if necessary.

The analysis of the results was performed according to the intention to treat (ITT).

The primary endpoint was to achieve normalization of fecal calprotectin level (i.e., result < 250.00 µg/g) after 12 weeks of nutritional therapy

Secondary endpoints were defined as obtaining at week 12:The normalization of fecal calprotectin level with normal CRP and PCDAI scores;A 50% decrease in fecal calprotectin level compared to baseline;The statistically significant decrease in PCDAI score and inflammatory parameters (fecal calprotectin, CRP and ESR) after 12 weeks of nutritional therapy in the entire studied population;Clinical remission (defined as PCDAI score <10.00 points) and clinical response (defined as a decrease in PCDAI score of at least 12.50 points or a score <10.00 points) in patients who were not in clinical remission at baseline (PCDAI ≥ 10.00);Normalization of fecal calprotectin level (i.e., score < 250.00 µg/g) in patients who were (PCDAI < 10.00) and who were not in clinical remission at baseline (PCDAI ≥ 10.00).

Statistical analysis of the results obtained after 12 weeks of nutritional therapy was performed using the Wilcoxon paired observational test. Results with *p* < 0.05 were considered significant.

## 3. Results

Normalization of fecal calprotectin level was achieved in 17/48 (35.42%) of children and a decrease of at least 50% in 27/48 (56.25%) of them. All 17 patients with normalization of fecal calprotectin at week 12 were in clinical remission, of which 9 (52.94%) were in remission at baseline and remained in remission throughout the study, and 8 (47.06%) achieved remission as a result of the nutritional intervention. Meanwhile, in the group of patients who were in clinical remission at week 12 (29/48), only slightly more than half (17/29; 58.62%) achieved normalization of the fecal calprotectin level at this stage of treatment.

Of the 17 patients who achieved normalization of fecal calprotectin level after 12 weeks of treatment, 16 (94.12%) had a normal CRP result (i.e., <0.50 mg/dL), of which 7 had normal result at baseline and 9 achieved this effect as a result of nutritional intervention. Overall, after 12 weeks of nutritional therapy, 16/48 patients (33.33%) had normalized fecal calprotectin level with normal CRP and PCDAI scores.

The decrease in fecal calprotectin, other inflammatory parameters (CRP and ESR) and PCDAI score between week 0 and week 12 were statistically significant (Table 2).

Of the 29 children who were not in clinical remission (PCDAI ≥ 10.00) at baseline, 16/29 (55.17%) achieved clinical remission (PCDAI < 10.00) and 20/29 (68.97%) achieved clinical response after 12 weeks of nutritional therapy (Figure 1). The decrease in PCDAI score and fecal calprotectin level between week 0 and week 12 in this group of patients was also statistically significant (Table 2). However, the percentage of patients who achieved a fecal calprotectin level < 250.00 ug/g was 27.60%, which was definitely lower than in the group of patients in clinical remission at baseline (50.00%) (Figure 2).

Median fecal calprotectin level was higher in the group of patients with baseline PCDAI ≥ 10.00, compared to the general population, as well as to the group initially in clinical remission. This trend continued even after the end of the 12-week cycle of nutritional therapy. However, a statistically significant decrease in fecal calprotectin level between weeks 0 and 12 was obtained in all 3 subgroups of patients (Figure 3).

Of the 48 patients included in the study, 44 (91.67%) attended a follow-up in week 6 and 37 (77.08%)—in week 12. The most common reason for premature termination of nutritional therapy was lack of therapeutic efficacy (6/11). In isolated cases, modification of therapy was necessary due to difficulties in adherence to the diet (2/11) or poor tolerance of the enteral formula (1/11). For 1 child, despite good efficacy, their parents made the decision to discontinue nutritional therapy after phase 1.

Among these patients, the alternative treatment implemented included: exclusive enteral nutrition (3/11); steroid therapy (6/11); and biological treatment (1/11).

One patient after the first phase of CDED + PEN completed with good effect was instructed on further dietary management and transferred to the care of an adult center.

Of the 13 patients with underweight/malnutrition at baseline, 9 (69.23%) completed the entire induction treatment cycle. All of them showed improvement in nutritional status, assessed by changes in the body mass index relative to centile charts, of which 5 showed an increase of 1 centile channel.

Throughout the 12-week period, there was no deterioration in nutritional status in any of the patients who received nutritional therapy.

## 4. Discussion

Dietary treatment with CDED + PEN is a relatively new treatment option in Crohn’s disease, hence the number of studies evaluating its effectiveness is small. To date, one publication is available on the pediatric population [32] and two on adult patients (34–35), but their methodology differs from that used in the presented study. In all of them, the main inclusion criterion was clinical disease activity assessed in terms of PCDAI/CDAI scores, and an elevated fecal calprotectin level was one of the additional criteria, not a prerequisite for inclusion in the study. However, according to the current ECCO/ESPGHAN 2020 guidelines for the treatment of Crohn’s disease, a score on the clinical disease activity scale alone should not be used as the main criterion for assessing the effect of therapy, as it is not a marker of mucosal inflammation. A recognized and non-invasive marker of mucosal healing is a fecal calprotectin level below 250.00 µg/g. Moreover, according to the available literature, in approximately half of the patients in clinical remission, mucosal inflammation is found to persist. Therefore, it is recommended that fecal calprotectin level should be the basis for therapeutic decisions, even in the absence of clinical symptoms of the disease [25]. According to these guidelines, the increased fecal calprotectin level (≥250.00 µg/g) was considered the main criterion for inclusion in our study, unlike in previous published reports. Changes in this parameter provided the basis for evaluating therapeutic effects. Presentation of results in such a way is more objective and enables assessment of the full effect of therapy, i.e., achievement of deep remission along with mucosal healing.

The gold standard for noninvasive assessment of mucosal healing and, thus, the effects of CD treatment is now considered to be the joint analysis of three parameters, i.e., fecal calprotectin level, CRP, and clinical disease activity [25]. In the patients under analysis, a 12-week treatment with CDED + PEN resulted in normalization of fecal calprotectin level in 35%, while 33% achieved such an effect in combination with normal CRP and PCDAI scores. Due to the lack of such data, it is not possible to compare these results with three previous publications concerning this treatment modality [32,34,35]. In the data from the literature on pharmacological treatment, the percentage of patients achieving mucosal healing after the period of induction therapy was several percent for steroids [36] and ranged from several to nearly 40% for biological treatment, according to the molecule used [37]. The findings of this study, especially when taking into account the side effects of corticosteroids, the high cost of biological therapy, and its limited availability in Poland, should be considered as very promising.

The decrease in fecal calprotectin level in the studied group was statistically significant, as in previous publications [32,34,35]. Moreover, the normalization of FCP level achieved 35% of patients and more than half of them obtained a minimum 50% reduction in this parameter. Therefore, it can be concluded that treatment with CDED + PEN can be effective in more than a third of the population of patients with CD. It may be reasonable to consider prolongation of the induction stage of CDED + PEN in those whose FCP level have not been normalized within 12 weeks of nutritional treatment but achieved improvement in this respect and obtained other benefits from the therapy (e.g., achieved clinical remission or normalization of other inflammatory parameters). Experience with pharmacology reveals that some CD patients require a longer duration of a therapy to achieve full remission.

Baseline median fecal calprotectin level was higher in patients with clinical manifestation of Crohn’s disease than in those in clinical remission. The same observation applied to therapeutic effects—the percentage of patients who achieved normalization of this parameter after 12 weeks of nutritional therapy was almost halved in patients with clinical remission at baseline compared with a quarter of children who showed clinical signs of CD exacerbation (50.00% vs. 27.60%, respectively). These results confirm the validity of making therapeutic decisions based on the evaluation of the fecal calprotectin level, the rise of which is often the first symptom of disease exacerbation that precedes clinical symptoms. Based on the results in our group of patients, it can be proved that clinical symptoms of disease are associated with more advanced mucosal inflammation, and treatment modification only at this stage is associated with poorer prognosis.

The percentage of patients who achieved clinical remission in the studied population was 55.2%, which was lower compared to previous published reports (75.6%, Levine 2019; 82.1%, Szczubełek 2021). A plausible explanation for the slightly worse efficacy among our patients is that an objective marker of active inflammation, i.e., fecal calprotectin level of ≥250.00 µg/g, was a condition for inclusion in the study. In the above-mentioned two publications, the primary criterion for inclusion in the analysis was the clinical manifestation of the disease, which was assessed based on the disease activity score (PCDAI/CDAI), i.e., partially subjective interpretation of symptoms by the patient. An elevated score on the clinical disease activity scale does not necessarily correlate with objective markers of mucosal inflammation and, thus, not indicate an actual exacerbation of the disease.

Tolerability of therapy was high (96%) and only two patients discontinued CDED + PEN due to difficulty with dietary compliance. This result is comparable to the previous report on the pediatric population (Levine 2019; 97.5%) and is slightly better compared to the result obtained in adult patients (Szubelek 2021, 87%). As reported in previous publications, nutritional therapy is generally less well tolerated among adult CD patients than in the pediatric population [38]. However, all three publications reveal that the ability of the patients to adhere to the CDED + PEN diet is very high and is not a barrier to the use of this treatment modality.

Levine’s 2019 publication, which was the only one concerning the pediatric population, did not analyze the effect of CDED + PEN on nutritional status based on the assessment of BMI scores. The authors only showed that the body weight of patients increased significantly after 6 weeks of nutritional treatment. In our group of patients, all of those with a baseline malnutrition/weight deficiency and in whom nutritional therapy was effective had improved nutritional status. This result proves the added benefit of nutritional therapy over pharmacological therapy in the pediatric population. Moreover, none of the patients who completed a 12-week treatment had nutritional deterioration, which proves that, despite severe restrictions, the combination of CDED and partial enteral nutrition makes this nutritional therapy safe and does not pose a risk of worsening nutritional status.

The main limitation of this study is its observational, open-label nature and the lack of a control group. In view of the efficacy results of CDED + PEN, which were published by Levine in 2019, suggesting alternatively more difficult to apply and less effective, exclusive enteral nutrition was not appropriate for our patients.

Another limitation is that the patients did not have follow-up endoscopy after 12 weeks of treatment. Such management, however, is not used in daily practice and, according to current guidelines, the authoritative marker of mucosal healing is the measurement of fecal calprotectin level. This parameter was used in this analysis as a basis for evaluating therapeutic effects.

It might be interesting to analyze the efficacy of CDED + PEN, according to whether the nutritional therapy was the first treatment or the patients had already been treated with pharmacotherapy. Due to the small group size, providing such a comparison was not possible in this publication. The authors plan to provide such a summary when sufficient group size is available for analysis.

In spite of these limitations, the most important aspect of this publication that brings new conclusions is the presentation of results based on the analysis of changes in an objective marker of mucosal inflammation, i.e., fecal calprotectin level. This helps to evaluate the full therapeutic effect of CDED + PEN, which is the achievement of deep remission along with mucosal healing.

## 5. Conclusions

In summary, we can conclude a 12-week course of induction therapy with CDED + PEN has a beneficial effects on fecal calprotectin level in children with Crohn’s disease and increased level of this parameter. In the group under analysis, this nutritional intervention resulted in a significant decrease in FCP level and contributed to normalization of this parameter in one-third of patients, which is a similar or better result compared to pharmacological treatment. We can also state that in patients with baseline clinical manifestation of Crohn’s disease the mucosal inflammation is more advanced and the therapeutic effect of CDED + PEN may be worse than in those who are initially in clinical remission. Finally, in our opinion extending the induction period of CDED + PEN is worth considering in patients who benefit from it but do not achieve a full effect after 12 weeks. The results presented in this publication provide further evidence of the good efficacy of CDED + PEN in the treatment of active Crohn’s disease in children and thus another step towards changing the recommendation of nutritional therapy in CD.

## Figures and Tables

**Figure 1 jcm-11-04146-f001:**
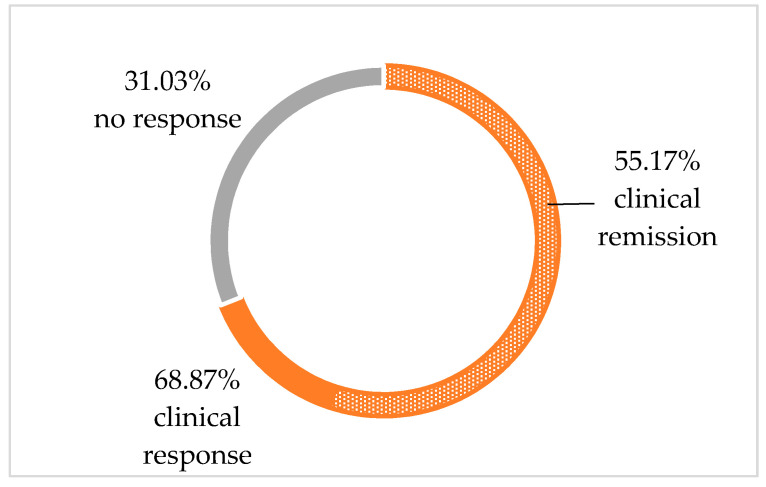
Therapeutic effect after week 12 of nutritional intervention in patients with baseline PCDAI ≥ 10.00 (assessment of clinical manifestation of the disease based on the PCDAI score).

**Figure 2 jcm-11-04146-f002:**
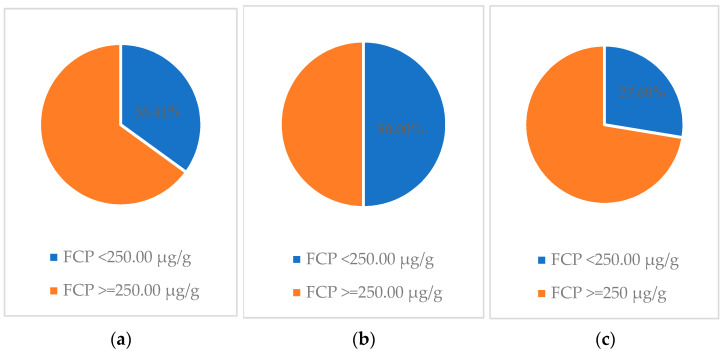
Percentage of patients in whom normalization of fecal calprotectin level after 12 weeks of nutritional therapy was achieved: (**a**) total population; (**b**) patient population with baseline PCDAI <10.00; and (**c**) population with baseline PCDAI ≥10.00.

**Figure 3 jcm-11-04146-f003:**
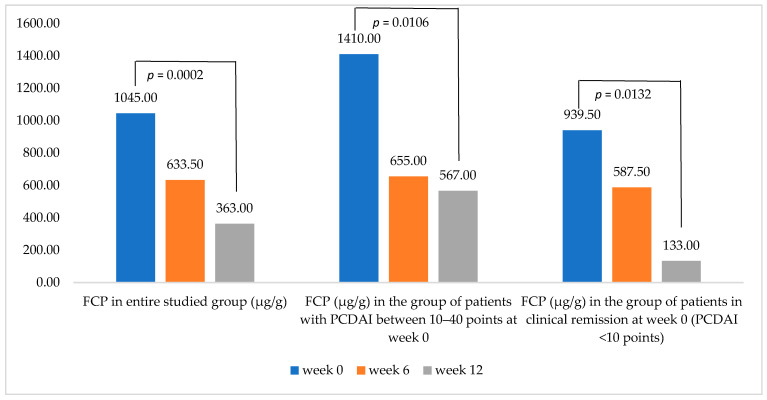
Comparison of changes in fecal calprotectin level at follow-up visits, depending on baseline clinical disease activity (i.e., PCDAI score).

**Table 1 jcm-11-04146-t001:** Baseline characteristics of the studied group.

Week 0	Total Population (*n* = 48)
Male	27 (56.00%)
Age at diagnosis, years	From 3 to 16 years (median = 11.98; IQR = 4.78)
Age at week 0, years	From 4 to 17 years (median = 13.43; IQR = 4.00)
Patients with a new diagnosis of the disease	23 (48.00%)
Duration of disease before initiation of nutritional therapy in the group with subsequent exacerbation (N =), years	From 3 months to 7 years (median = 1.21; IGR = 2.35)
Severity of clinical symptoms of disease (baseline PCDAI score):	
Clinical remission (0.00–7.50) M	17 (35.42%)
Mild (10.00–27.50)	27 (56.25%)
Average (30.00–40.00)	3 (6.25%)
No data	1 (2.08%)
Nutritional status (BMI kg/m^2^):	
Underweight = BMI 3–15 pp	7 (14.58%)
Malnutrition = BMI < 3 pp	6 (12.50%)
Total underweight + malnutrition	13 (27.08%)

PCDAI—pediatric Crohn’s disease activity index; pp—percentile; IQR—interquartile range.

**Table 2 jcm-11-04146-t002:** Analysis of changes in disease activity parameters at follow-up visits.

Parameter	Baseline (Week 0)	1st Follow-Up (Week 6)	2nd Follow-Up (Week 12)	*p*-Value (2nd Follow-Up vs. Week 0)
**Entire studied population (*n* = 48)**				
Fecal calprotectin level (µg/g)	1045.00 (IQR = 1188.00)	633.50 (IQR = 1268.00)	363.00 (IQR = 665.00)	0.0002
PCDAI (pt)	12.50 (IQR = 17.50)	5.00 (IQR = 7.5)	5.00 (IQR = 7.50)	0.0002
CRP (mg/dL)	1.00 (IQR = 1.78)	0.20 (IQR = 0.42)	0.19(IQR = 0.25)	0.0002
ESR (mm/h)	21.00 (IQR = 19.50)	10.00 (IQR = 13.00)	11.00 (IQR = 11.00)	0.0014
**Population of patients who were not in clinical remission at baseline, PCDAI ≥ 10 (*n* = 29)**				
Fecal calprotectin level (µg/g)	1410.00 (IQR = 1839.00)	655.00 (IQR = 1919.50)	567.00 (IQR = 1147.50)	0.0106
PCDAI (pt)	20.00 (IQR = 7.50)	7.50 (IQR = 5.00)	5.00 (IQR = 5.00)	0.0002
**Population of patients who were in clinical remission at baseline, PCDAI < 10 (*n* = 18)**				
Fecal calprotectin level (µg/g)	939.50 (IQR = 989.00)	587.50 (IQR = 1433.50)	133.00 (IQR = 284.50)	0.0132

Results expressed as a median. IQR—interquartile range.

## Data Availability

https://mymodulife.com/experts/ (accessed on 16 Jul 2022).

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
