# Peer review of "Effect of the Crohn’s Disease Exclusion Diet (CDED) on the Fecal Calprotectin Level in Children with Active Crohn’s Disease"

_jcm, 2022, doi:10.3390/jcm11144146_

Round 1

Reviewer 1 Report

Thank you for the opportunity to review this nice dietary intervention study which adds to the current evidence base for using CDED+PEN as remission induction and/or bridging therapy for active luminal Crohn's disease in the Polish paediatric population. This study examines the feasibility of using faecal calprotectin (FC) as a objective marker of clinical response/remission in the absence of endoscopy to assess mucosal healing.

I have some questions about the accuracy and interpretability of the data for the authors:

1. Why did you use FC 250ug/g as a cut off value for your primary end point - there is no citation and it differs from the Levine 2019 manuscript which is heavily referred to

2. Can you please define the different PCDAI thresholds for your secondary outcomes and use citations

3. Which malnutrition assessment tool did you use beyond BMI (BMI is not a diagnostic tool for malnutrition)

4. Did you measure adherence and compliance to therapy and if so, which tools did you use and do you have the data to support this?

5. What did you use to measure tolerability of therapy - please define this in methodology

6. For accuracy, it would be helpful to standardise the terminology for CDED+PEN and the phases and the mandatory foods etc as readers will be familiar with the original RCT. This is also relevant where PEN is referred to complete enteral nutrition - the correct terminology is partial enteral nutrition (PEN) or exclusive enteral nutrition (EEN)

7. It is unclear to me why there is so much emphasis in the result section on the subgroup who had PCDAI <10 at baseline? I understand this is clinical remission before even starting the diet therapy but in your statistics section, comparing PCDAI and FC is not an endpoint nor is it a subgroup analysis. The data is difficult to follow at times and Figure 2 does not increase clarity around this. 

With regards to structure and language, the following should be addressed:

1. The introduction is lengthy and could be distilled for accuracy and readability - references also require review for more recent

2. presentation of data to 3 significant figures throughout the manuscript and tables

3. Review use of English language and consistency in the way data is presented and terminology is used. For example with data, line 192-198 of results, the number of participants and the % is expressed in 3 different ways. Typically it would be xxx/48 (xx%)

4. Some results are presented in methods/materials section - please amend

5. The discussion requires review for accuracy

Author Response

Dear Reviewer,

Thank you very much for all your valuable comments, which will certainly improve the quality of our publication.

Please find attached our answers.

Best regards

Malgorzata Matuszczyk

Reviewer 2 Report

The present study on CDED and calprotectin levels is of great interest and provides more much-needed data regarding effectiveness of different nutritional therapies in pediatric Crohn's disease.

Nevertheless, there are some points that could be improved:

1. The results are not presented clearly for me, since results of week 0 and 12 were mentioned and compared for the most part, but then there are tables and figures with results from week 6. Since the response to phase I of the CDED is also relevant, I would include the week 6 results in the abstract and other parts of the paper as well.

2. You stated that elevated fCP was the main reason for inclusion of the patients in the study. Where those only first diagnosed children? And if not, how did you convince subjects who were in clinical remission to do a 12-week nutritional therapy? Please clarify

3. Any information about concomidant therapy is missing.

4. CD-TREAT is the other diet containing solid foods. Please add this to the introduction and discussion.

5. Please add and discuss the following recently published paper: Ghiboub M, Penny S, Verburgt CM, Sigall Boneh R, Wine E, Cohen A, Dunn KA, Pinto DM, Benninga MA, de Jonge WJ, Levine A, Van Limbergen JE. Metabolome changes with diet-induced remission in pediatric Crohn's disease. Gastroenterology. 2022 Jun 6:S0016-5085(22)00596-0. doi: 10.1053/j.gastro.2022.05.050. Epub ahead of print. PMID: 35679949.

6. Did all subjects receive Modulen or were there exceptions as you mentioned. Please provide the exact data?

7. Since severe courses were excluded, I would add to the title that it is a “mild-moderate” CD

8. Line 47-48: I don´t fully agree with that. There is an armamentarium of good new therapies that are being tested or are already available. I would rephrase this a bit

Minor comments:

Line 20: Explain abbreviation “ITT”

Line 80: Glucocorticosteroids

Line 99 remove “the”

Line 117 remove “the”

Author Response

Dear Reviewer,

Thank you very much for all your valuable comments, which will certainly improve the quality of our publication.

Please find attached our answers.

Best regards

Malgorzata Matuszczyk

This manuscript is a resubmission of an earlier submission. The following is a list of the peer review reports and author responses from that submission.